# Pravastatin Alleviates Radiation Proctitis by Regulating Thrombomodulin in Irradiated Endothelial Cells

**DOI:** 10.3390/ijms21051897

**Published:** 2020-03-10

**Authors:** Hyosun Jang, Seo-Young Kwak, Sunhoo Park, Kyuchang Kim, Young-heon Kim, Jiyoung Na, Hyewon Kim, Won-Suk Jang, Sun-Joo Lee, Min Jung Kim, Jae Kyung Myung, Sehwan Shim

**Affiliations:** 1Laboratory of Radiation Exposure & Therapeutics, National Radiation Emergency Medical Center, Korea Institute of Radiological and Medical Sciences, Seoul 01812, Korea; hsjang@kirams.re.kr (H.J.); ksy@kirams.re.kr (S.-Y.K.); sunhoo@kirams.re.kr (S.P.); kimkyuch@gmail.com (K.K.); bumbroberto1@nate.com (Y.-h.K.); ekfrifgksk@gmail.com (J.N.); hw0227@kirams.re.kr (H.K.); wsjang@kirams.re.kr (W.-S.J.); sjlee@kirams.re.kr (S.-J.L.); kimmj74@kirams.re.kr (M.J.K.); tontos016@naver.com (J.K.M.); 2Department of Pathology, Korea Cancer Center Hospital, Korea Institute of Radiological and Medical Sciences, Seoul 01812, Korea

**Keywords:** pravastatin, radiation, proctitis, thrombomodulin

## Abstract

Although radiotherapy plays a crucial in the management of pelvic tumors, its toxicity on surrounding healthy tissues such as the small intestine, colon, and rectum is one of the major limitations associated with its use. In particular, proctitis is a major clinical complication of pelvic radiotherapy. Recent evidence suggests that endothelial injury significantly affects the initiation of radiation-induced inflammation. The damaged endothelial cells accelerate immune cell recruitment by activating the expression of endothelial adhesive molecules, which participate in the development of tissue damage. Pravastatin, a cholesterol lowering drug, exerts persistent anti-inflammatory and anti-thrombotic effects on irradiated endothelial cells and inhibits the interaction of leukocytes and damaged endothelial cells. Here, we aimed to investigate the effects of pravastatin on radiation-induced endothelial damage in human umbilical vein endothelial cell and a murine proctitis model. Pravastatin attenuated epithelial damage and inflammatory response in irradiated colorectal lesions. In particular, pravastatin improved radiation-induced endothelial damage by regulating thrombomodulin (TM) expression. In addition, exogenous TM inhibited leukocyte adhesion to the irradiated endothelial cells. Thus, pravastatin can inhibit endothelial damage by inducing TM, thereby alleviating radiation proctitis. Therefore, we suggest that pharmacological modulation of endothelial TM may limit intestinal inflammation after irradiation.

## 1. Introduction

Although radiation therapy is an important part of cancer treatment, with approximately 50% of all cancer patients receiving radiation therapy during treatment, it induces acute and chronic inflammation in healthy organs [1]. In particular, radiation proctitis is one of the most common side effects in patients receiving radiation therapy for intra-pelvic cancer such as prostate and cervical cancer [2,3]. Acute radiation proctitis occurring in 50–78% patients is clinically associated with diarrhea, defecation abnormality, and anal pain [4,5], with histopathological damage such as mucosal erosion, inflammatory cell infiltration, and crypt abscesses [6]. Chronic radiation proctitis occurring in approximately 80% patients is associated with hemorrhage, with stenosis occurring in approximately 1% and fistula formation in 0.4% patients [4,5,7]. Histopathological changes, such as damage throughout the gut well, vascular dystrophy, and uncontrolled scarring leading to tissue fibrosis have also been observed [4,8]. Thus, radiation proctitis decreases the quality of life of patients receiving radiation therapy, who often require either interruption of therapy or other modifications that may forestall the optimal completion of the original treatment plan.

Vascular abnormality is consistently observed in healthy tissue of patients undergoing radiation exposure, and endothelial damage has been described as a crucial event in the initiation and progression of the side effects of radiation in normal tissue [1,9,10]. Endothelial cells mediate the recruitment and migration of inflammatory leukocytes to areas of tissue damage and infection via sustained expression of endothelial adhesion molecules, chemokine production, and localized or disseminated tissue dysfunction [11,12,13,14,15,16].

Thrombomodulin (TM), a transmembrane glycoprotein in endothelial cells, plays an important role in maintaining normal endothelium function [17]. Generally, TM acts by forming a complex with thrombin, thereby inhibiting coagulation. The absence of TM in endothelial cells is associated with several diseases, including vascular disease [18], graft-versus-host disease [19], and infectious diseases [20]. Therapy with recombinant TM or related molecules increases the survival rate of patients with disseminated intravascular coagulation and severe sepsis [21,22,23]. Although decrease in TM has been observed in radiation-induced proctitis and a bladder injury model [24,25], therapeutic studies targeting TM in radiation injury are limited.

Commonly known as statins, 3-Hydroxyl-3-methyl coenzyme A reductase inhibitors are effective in lowering plasma low density lipoprotein-cholesterol concentration. Interestingly, statins have clinically beneficial pleiotropic effects unrelated to their lipid-lowering effects. Pravastatin, one of the stains, has been shown to improve endothelial dysfunction, enhance the stability of atherosclerotic plaques, and decrease oxidative stress, coagulation, adhesion of leukocytes, and vascular inflammation. In addition, we have previously reported that pravastatin alleviated radiation-induced epithelial damage in the small intestine by reducing oxidative stress [26].

In this study, we investigated the therapeutic effects of pravastatin on radiation proctitis via regulation of endothelial TM. We showed that the epithelium of the crypts is damaged in response to radiation toxicity; however, application of pravastatin attenuated rectal epithelium damage. Activation of TM by pravastatin resulted in the recovery of endothelial dysfunction and decreased inflammatory response. These results suggested that pravastatin can protect from radiation proctitis by upregulating TM expression in endothelial cells, thereby attenuating inflammatory response and improving epithelial damage.

## 2. Results

### 2.1. Pravastatin Inhibited Inflammation Response in Endothelial Cells by Regulating TM

Endothelial cells play key roles in mucosal immune homeostasis, acting as gatekeepers that prevent leukocytes from migrating from the intravascular to the interstitial space. Irradiation facilitates the recruitment of circulating monocytes by inducing the expression of intercellular adhesion molecule-1 (ICAM-1) and vascular cell adhesion molecule-1 (VCAM-1) on endothelial cells [27]. We also observed that radiation exposure impairs endothelial function in human umbilical vein endothelial cells (HUVECs) based on tube formation assays (Figure 1a–c). Furthermore, compared to that in the IR group, pravastatin treatment significantly increased total tube length and the number of branch points in irradiated HUVECs (Figure 1a–c). To investigate the effect of pravastatin on radiation-induced endothelial dysfunction, we performed the leukocyte adhesion assay, which assesses the attachment of inflammatory cells to damaged endothelial cells progressing to inflammation. The number of endothelial-attached green-positive cells (THP-1) was higher in the IR group than in the control group; furthermore, compared to that in the IR group, pravastatin treatment decreased the number of endothelial-attached THP-1 cells (Figure 1d). TM, an anti-inflammatory molecule, plays a particularly important role in maintaining normal endothelial cell function. TM blocks leukocyte adhesion to the activated endothelium, and mouse with TM malfunction shows increase in leukocyte influx in several models of inflammation [28]. The irradiated HUVECs showed reduction in the expression of TM and elevation in plasminogen activator inhibitor-1 (PAI-1), ICAM-1, and VCAM-1 expression, which are regulated by TM (Figure 1e–h). PAI-1 is a well-known serine protease inhibitor and anti-fibrolytic molecule, which is inactivated by TM. Patients with radiation-induced intestinal injury display increased PAI-1 levels [29], and a mouse model of genetic PAI-1 deficiency shows inhibition of radiation-induced intestinal damage [30]. Otherwise, pravastatin treatment improved TM levels in the irradiated endothelial cells and decreased the levels of endothelial adhesion molecules (Figure 1e–h). Taken together, pravastatin attenuated radiation-induced endothelial dysfunction by regulating TM.

### 2.2. TM Improved Radiation-Induced Endothelial Dysfunction

To identify the effects of TM in radiation-induced endothelial dysfunction, we performed tube formation and leukocyte adhesion assay using recombinant TM. In the tube formation assay, TM treatment of the irradiated HUVECs improved branch point number and tube length (Figure 2a–c). In addition, leukocyte adhesion was higher in the IR group than in the control, whereas TM treatment inhibited the attachment of THP-1 cells to the irradiated HUVECs (Figure 2d). Compared to those in the IR group, the expression of *PAI-1*, *ICAM-1*, and *VCAM-1* were significantly suppressed in TM-treated irradiated HUVECs (Figure 2e–g). Taken together, TM inhibited leukocyte adhesion to the irradiated endothelial cells and suppressed the expression of endothelial adhesion molecules.

### 2.3. Pravastatin Mitigated Radiation Proctitis

To investigate the effects of pravastatin on radiation proctitis, we performed localized irradiation on the colorectum of female mice. We analyzed the histopathological changes in the colorectal lesion at two and four weeks after radiation exposure using hematoxylin and eosin (H & E) staining. In irradiated (IR) mice, remarkable crypt destruction with edema, crypt abscess, irregular epithelial cell, and inflammatory cell infiltration in the mucosa were observed in the colorectum (Figure 3a). However, compared to that in the IR group, pravastatin-treated IR mice showed marked rescue of crypt damage (Figure 3a). Also, histological score was significantly alleviated in the pravastatin-treated IR group compared to that in IR group (Figure 3a,d). Next, we performed periodic acid-Schiff base (PAS) staining and immunostaining of claudin 3, a tight junction protein, and analyzed plasma diamine oxide (DAO) level in the pravastatin-treated IR mice to identify the effect of pravastatin on radiation-induced epithelial damage. The purple colored cells after PAS staining indicated the goblet cells, which protect the epithelium by producing mucins [31]. Claudin 3 is involved in intestinal epithelial barrier function and sensitivity to radiation exposure [32]. The number of purple colored cells in the PAS staining was markedly lower in the IR group than in the control group (Figure 3b). Pravastatin treatment improved goblet cell damage in the irradiated colorectum (Figure 3b) and alleviated claudin 3 expression in the irradiated colorectal lesions (Figure 3c). Increase in plasma DAO levels is indicative of epithelial damage [33]. Compared to those in the control group, radiation exposure of colorectal lesions consistently increased the levels of plasma DAO at two and four weeks (Figure 3e). The pravastatin-treated IR group showed significantly lower plasma DAO levels than the IR group (Figure 3e). As plasma C-reactive protein (CRP) is a marker of systemic inflammation [27], the IR group showed markedly higher levels of plasma CRP than the control group at two and four weeks (Figure 3f). The plasma CRP levels were significantly lower in the pravastatin-treated IR group than in the IR group at four weeks (Figure 3f). Therefore, pravastatin mitigated radiation-induced colorectal injury, including those to the epithelium, in a radiation proctitis model.

### 2.4. Pravastatin Accelerated TM Expression in Radiation-Induced Proctitis

Irradiated HUVECs showed functional defect with decreased TM expression in our in vitro data. Pravastatin treatment alleviated endothelial dysfunction by increasing TM expression in irradiated HUVECs. Next, we investigated whether the therapeutic effects of pravastatin on radiation proctitis is associated with TM. TM was expressed in the endothelial cells of the surrounding crypts in the healthy colorectal tissue (Figure 4a,b). Similar to the results of the in vitro experiment, the protein and mRNA levels of TM were markedly lower in the IR mice than in the control mice. However, compared to that in the IR mice, pravastatin treatment increased TM levels at two and four weeks. The mRNA levels of *Pai-1*, *Icam-1*, and *Vcam-1* were higher in the irradiated colorectum than in the control group (Figure 4c,d). Pravastatin treatment significantly decreased the *Pai-1*, *Icam-1*, and *Vcam-1* mRNA levels compared to that in the IR group at two and four weeks (Figure 4c,d). Thus, pravastatin regulates TM and inhibits the expression of endothelial adherent molecules.

### 2.5. Pravastatin Attenuated Inflammatory Response in Radiation-induced Colorectal Damage

Leukocyte infiltration in the inflamed tissue is one of the important markers of the progress of inflammation. TM blocks leukocyte adhesion to the activated endothelium and inhibit leukocyte influx in injured tissue [28]. Neutrophil and macrophage accumulation were observed in the mucosal layer of the irradiated colorectum at two and four weeks (Figure 5a,b). The epithelial layer of the severely inflamed lesions of the IR group was infiltrated by neutrophils (observed using myeloperoxidase immunostaining) and macrophages (CD68 immunostaining) (Figure 5a,b). However, pravastatin treatment reduced neutrophil and macrophage infiltration in the mucosal lesion of the irradiated colorectum (Figure 5a,b). We also identified that the colorectal tissue of the IR group showed upregulation of inflammatory cytokines and chem okines, such as interleukin 1β (Il1β), chemokine ligand 1 (Cxcl1), tumor necrosis factor α (Tnfα), and monocyte chemoattractant protein 1 (Mcp1) (Figure 5c–f). Compared to the IR group, pravastatin treatment reduced the expression of these inflammatory cytokines and chemokines at two and four weeks (Figure 5c–f). Taken together, pravastatin attenuated inflammatory response by blocking the infiltration of neutrophils and macrophages in radiation proctitis.

## 3. Discussion

Although radiotherapy plays a crucial in the management pelvic tumors, the major limitation of this treatment is its toxicity on surrounding healthy tissues such as the small intestine, colon, and rectum. In particular, proctitis is a major clinical complication of pelvic radiotherapy, and has been reported to occur in 5–20% of patients who have had radiation therapy in the pelvic region [34]. Radiation exposure to colorectal lesions induces mucosal inflammation and epithelial damage with loss of epithelial stem cells, and increases mucosal permeability and gut pathogen infiltration, which exacerbate mucosal inflammation [35]. Persistent colorectal inflammation induces bowel well hypertrophy, with vascular dystrophy and uncontrolled scarring leading to tissue fibrosis [4,8]. Colorectal complications affect the quality of life of patients for several years after the end of the radiotherapy. However, management of patients suffering from radiotherapy-related proctitis is limited to symptomatic treatments. For investigating therapeutic agents suitable for patients of radiation proctitis, we established a mouse model of radiation proctitis via the localized irradiation of the colorectum. In our study, the irradiated mouse showed histological damage and continuous inflammation in the colorectal lesion, with increased plasma CRP levels, which are similar to that observed in patients with radiation proctitis [6,36].

Radiation-induced endothelial injury has been described as a crucial event in the initiation and progression of inflammation in normal tissue [10]. As endothelial cells act as gatekeepers that prevent leukocytes from migrating from the intravascular to the interstitial space, damaged endothelial cells accelerate immune cell recruitment by activating the expression of endothelial adhesive molecules, which participate in the development of tissue damage. Therefore, protection from endothelial damage after radiation exposure with pharmacological agents may limit one of the crucial steps contributing to the pathogenesis of this condition.

Statins are effective in lowering the plasma low density lipoprotein-cholesterol concentration. Interestingly, many of these drugs have clinically beneficial pleiotropic effects that are not related to their lipid-lowering effects. As the vascular endothelium may be a major target for the pleiotropic effects of statins, application of statins has been shown to improve endothelial dysfunction, enhance the stability of atherosclerotic plaques, and decrease oxidative stress, coagulation, and vascular inflammation. Furthermore, pravastatin exerts persistent anti-inflammatory and anti-thrombotic effects on irradiated endothelial cells and inhibits interactions of leukocytes and damaged endothelial cells [37,38]. Holler et al. (2009) demonstrated that pravastatin exerts a therapeutic effect on radiation dermatitis and abolishes radiation-induced vascular activation by decreasing interactions between leukocytes and the endothelium in vivo [38]. However, studies of pravastatin’s effects on radiation proctitis-associated endothelial damage are limited.

In this study, we observed that pravastatin can attenuate radiation proctitis by improving endothelial dysfunction. Radiation induces crypt destruction with abscess formation, inflammatory cell infiltration, and epithelial damage, including reduction in tight junction and increase in plasma DAO levels. In addition, irradiated HUVECs lose angiogenetic property and show acceleration in the attachment of leukocytes, with an increased expression of endothelial adhesion molecules. Furthermore, radiation induced the infiltration of inflammatory cells such as neutrophils and macrophages, and upregulated inflammatory cytokines by increasing the expression of endothelial adhesion molecules in the colorectum. In contrast, the pravastatin-treated proctitis group showed alleviation of histological injury with reduction in histological score and epithelial damage. As pravastatin suppressed endothelial attachment molecules and leukocyte adhesion to the irradiated endothelial cells, leukocyte infiltration was inhibited in the irradiated colorectal lesion. Therefore, we concluded that pravastatin attenuated radiation proctitis and improved radiation-induced endothelial damage.

TM is well-known as an anti-inflammatory molecule as it inhibits leukocyte adhesion to the activated endothelium. A mouse model lacking TM shows increased leukocyte influx with severe inflammation [28,39]. Ionizing radiation also interrupts TM expression and function. For example, radiation-induced inflammatory cytokine (TNFα) or protease (granulocyte elastase) suppress TM transcription or cleave TM from the endothelial cell membrane to pass it into circulation [40,41]. Furthermore, ionizing radiation directly inactivates TM by oxidation of a specific methionine on a part of TM [42]. Loss of TM appears to be involved in both early and delayed radiation toxicity [43,44]. Geiger et al. (2012) reported that application of exogenous TM protects mice from death by accelerating recovery of hematopoietic cells [45]. In this study, we observed that pravastatin treatment upregulated TM in the irradiated colorectal tissue and endothelial cells, thereby alleviating radiation proctitis. 

The addition of exogenous TM assisted in the recovery from radiation-induced anti-angiogenesis and inhibited leukocyte adhesion to the damaged endothelial cell by suppressing PAI-1, ICAM-1, and VCAM-1. These results indicated that TM is a therapeutic target for radiation-induced endothelial injury.

In summary, the present study revealed the therapeutic effects of pravastatin on radiation-induced endothelial injury both in vitro and in vivo. Our findings highlight the pivotal role of TM on the endothelium during radiation-induced toxicity. Although the precise mechanisms underlying activation of TM transcription by pravastatin are still unclear, the ability of pravastatin to suppress radiation-induced endothelial interaction with leukocytes is critical, as this results in inhibit of inflammation. Our results indicated that pravastatin inhibited endothelial activation by inducing TM, thereby alleviating radiation proctitis. Therefore, we suggest that pharmacological modulation of endothelial TM may limit intestinal inflammation after irradiation.

## 4. Materials and Methods

### 4.1. Cell Culture and Treatment

HUVECs (Lonza, Basel, Switzerland) at passages 5–8 were cultured in endothelial cell basal medium-2 (EBM-2; Lonza) supplemented with the endothelial cell growth medium-2 bullet kit (EGM-2; Lonza). Culture media were replaced every 3–4 days and the cultures were maintained in a humidified incubator at 37 °C in the presence of 5% CO_2_. The cells were irradiated with 10 Gy using a ^137^Cs γ-ray source (Atomic Energy of Canada, Chalk River, ON, Canada) at the rate of 3.81 Gy/min and then treated with 10 μM pravastatin (Sigma-Aldrich, St. Louis, MO, USA) and 100 ng/mL (R & D Systems) recombinant TM (rTM) within 1 h. THP-1 monocytes were maintained in Roswell Park Memorial Institute (RPMI)-1640 medium supplemented with 10% fetal bovine serum (FBS), 100 units/mL penicillin, and 100 µg/mL streptomycin at 37 °C in the presence of 5% CO_2_.

### 4.2. Capillary-Like Tube Formation Assay

The irradiated HUVECs were detached and re-plated in Matrigel (Corning, Corning, NY, USA)-coated 24-well culture plates. The culture medium was supplemented with EGM-2 with/without 10 µM pravastatin. After 4 h of incubation, the cells were fixed with 4% paraformaldehyde. Twelve microscopic fields per group were randomly selected to measure the total tube length and to count the number of branch points using the CellSens software (Olympus, Waltham, MA, USA).

### 4.3. Leukocyte Adhesion Assay

HUVECs at 90%∼95% confluence were irradiated and then treated with 10 μM pravastatin for 24 h. The THP-1 cells were labeled with CFSE (Life technologies) in PBS for 20 min. After extensive washing with RPMI 1640 medium containing 10% FBS, the labeled THP-1 cells were seeded at a density of 1.0 × 10^5^ cells/well into irradiated HUVECs treated with pravastatin, followed by incubation for 1 h at 37 °C. After incubation, the non-adherent THP-1 cells were removed via two gentle washes with PBS. Images were obtained at 485 nm excitation and 538 nm emission using a fluorescence microscope

### 4.4. RNA Extraction, Reverse Transcription-Polymerase Chain Reaction (RT–PCR), and Real-Time PCR Quantification

Harvested mouse colorectal tissues were immediately snap-frozen and stored at −80 °C until RNA extraction. Total RNA from colorectal tissue and HUVECs was isolated using the TRIzol reagent (Invitrogen, Carlsbad, CA, USA). cDNA was synthesized using the AccuPower RT premix (Bioneer, Daejeon, Korea) according to the manufacturer’s protocol. Real-time RT-PCR was performed using a LightCycler 480 system (Roche, San Francisco, CA, USA). The primer sequences are shown in Table 1. The expression levels of each target gene, determined using the LightCycler 480 system software (Roche), were normalized to that of mouse β-actin or human GAPDH. Cycle threshold values were used to calculate relative mRNA expression using the 2^−ΔΔCt^ method.

### 4.5. Mice

Specific pathogen-free (SPF) female C57BL/6 mice (7-week-old) were obtained from Harlan Laboratories (Indianapolis, IN, USA) and maintained under SPF conditions at the animal facility of the Korea Institute of Radiological and Medical Sciences (KIRAMS). All mice were housed in a temperature-controlled room with a 12 h/12 h-light/dark cycle, and food and water were provided *ad libitum*. The mice were acclimated for 1 week before experiments and assigned to the control (*n* = 10), irradiation (IR, *n* = 10), and irradiation with pravastatin treatment (IR+PS, *n* = 10) groups. All animal experiments were performed in accordance with the guidelines of and were approved by the Institutional Animal Care and Use Committee of KIRAMS.

### 4.6. Irradiation and Administration of Pravastatin

The colorectal regions of mice were irradiated with a single exposure to 27 Gy radiation (1 × 2 cm window center) at the rate of 2 Gy/min using an X-RAD 320 X-ray irradiator (Softex, Gyeonggi-do, Korea). After irradiation, mice were treated with a daily oral dose of 30 mg/kg/day pravastatin (Prastan^®^; Yungin Pharm, Seoul, Korea) for 4 weeks.

### 4.7. Histological Analysis

Colorectal samples of mice were fixed with a 10% neutral buffered formalin solution, embedded in paraffin wax, and sectioned longitudinally to a thickness of 4 µm. The sections were then stained with H & E and periodic acid-Schiff base (PAS). Evidence of intestinal mucosal injury was quantified (0 = none, 1 = mild, 2 = moderate, 3 = high) in the H & E-stained sections of the colorectum. The histological severity of radiation-induced colorectal damage was assessed from the degree of maintenance of epithelial architecture, crypt damage, vascular dilation, and infiltration of inflammatory cells in the lamina propria.

To perform immunohistochemical analysis, the slides were subjected to heat-induced antigen retrieval in Tris-EDTA buffer, pH 9.0, and then treated with 0.3% hydrogen peroxide in methyl alcohol for 20 min to block endogenous peroxidase activity. After three washes in phosphate-buffered saline (PBS), the sections were blocked with 10% normal goat serum (Vector ABC Elite kit; Vector Laboratories, Burlingame, CA, USA) and incubated with anti-claudin3 (Abcam, Cambridge, UK), anti-myeloperoxidase (Mpo; Abcam), anti-CD68 (Abcam), and anti-thrombomodulin (Tm; R&D) antibodies. After three washes in PBS, the sections were incubated with a horseradish peroxidase-conjugated secondary antibody (Dako, Carpinteria, CA, USA) for 60 min. The peroxidase reaction was developed using a diaminobenzidine substrate (Dako) prepared according to the manufacturer’s instructions, and the slides were counterstained with hematoxylin.

### 4.8. Measurements of Plasma DAO and CRP Levels

Mouse DAO (Cusabio, catalog number. CSB-E10090m, Wuhan, China) and CRP (R & D Systems, catalog number. MCRP00) levels were measured using a commercial enzyme-linked immunosorbent assay (ELISA) kit following the manufacturer’s protocol.

### 4.9. Statistical Analysis

Data are presented as the mean ± standard error of the mean (SEM). Statistical analyses were performed using one-way analysis of variance (ANOVA) with Tukey’s multiple comparison test. *P* values < 0.05 were considered statistically significant.

## Figures and Tables

**Figure 1 ijms-21-01897-f001:**
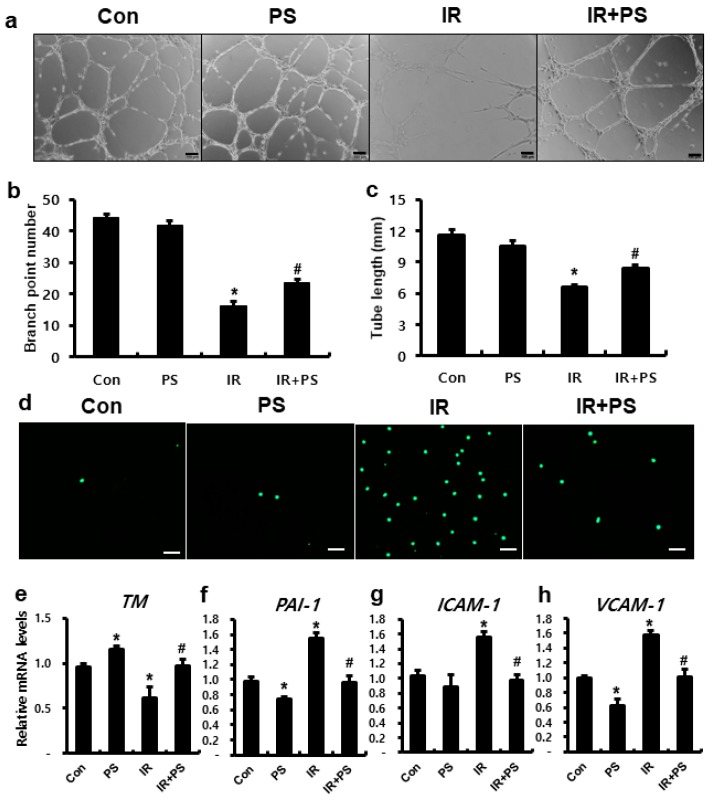
Pravastatin attenuated radiation-induced endothelial dysfunction with TM upregulation. (**a**) Capillary-like tube formation assays using human umbilical vein endothelial cells (HUVECs) in untreated (Con), pravastatin-treated (PS), irradiated (IR), and PS-treated IR (IR+PS) groups. Scale bar = 100 μm. (**b**) Total tube length and (**c**) the number of branch points were measured in each group. (**d**) Leukocyte adhesion assay using CSFE-labeled TPH-1 in pravastatin-treated IR HUVECs. Scale bar = 100 μm. mRNA levels of (**e**) *TM*, (**f**) *PAI-1*, (**g**) *ICAM-1*, and (**h**) *VCAM-1* in PS-treated IR HUVECs. Data are presented as the mean ± standard error of the mean; *n* = 3 for each group. ** P* < 0.05 compared to the control; *^#^ P* < 0.05 compared to the IR group.

**Figure 2 ijms-21-01897-f002:**
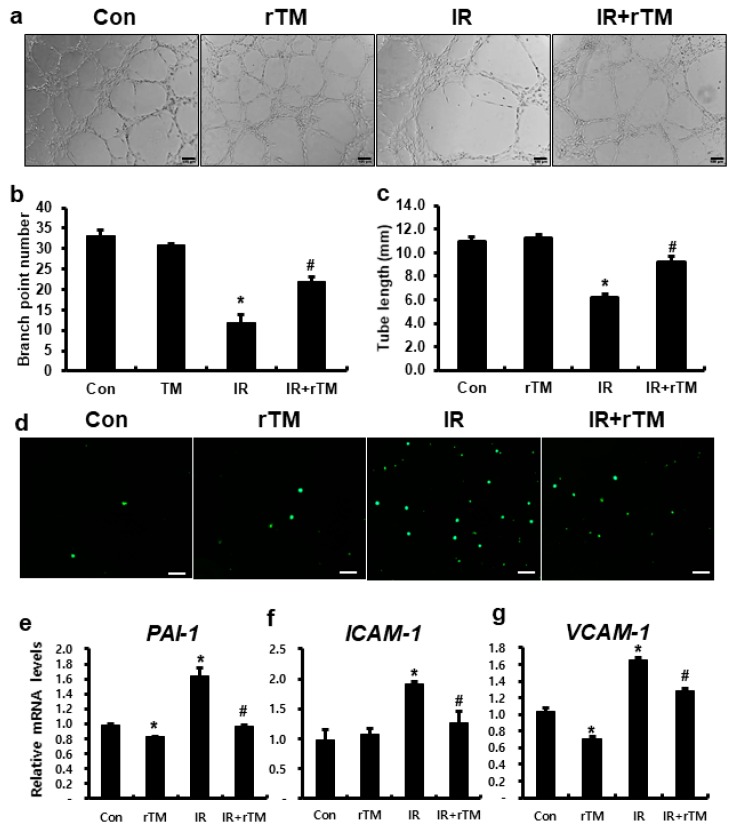
Thrombomodulin improved radiation-induced endothelial dysfunction. (**a**) Capillary-like tube formation assays using human umbilical vein endothelial cells (HUVECs) in untreated (Con), recombinant thrombomodulin-treated (rTM), irradiated (IR), and rTM-treated IR (IR+rTM) groups. Scale bar = 100 μm. (**b**) Total tube length and (**c**) the number of branch points were measured in each group. (**d**) Leukocyte adhesion assay using CSFE-labeled TPH-1 in Con, rTM, IR, and IR+rTM HUVECs. Scale bar = 100 μm. mRNA levels of (**e**) *TM*, (**f**) *PAI-1*, (**g**) *ICAM-1*, and (**h**) *VCAM-1* in rTM-treated IR HUVECs. Data are presented as the mean ± standard error of the mean; *n* = 3 for each group. ** P* < 0.05 compared to the control; *^#^ P* < 0.05 compared to the IR group.

**Figure 3 ijms-21-01897-f003:**
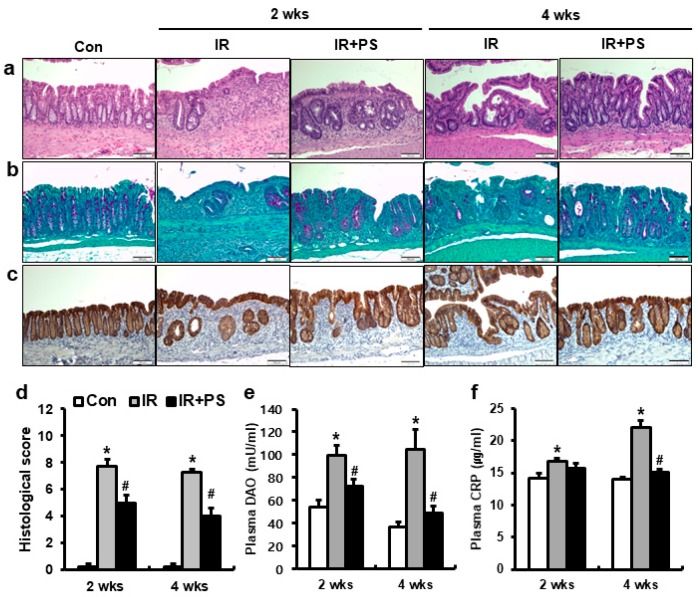
Pravastatin alleviated radiation proctitis by improving epithelial damage. (**a**) Hematoxylin-eosin (H&E), (**b**) periodic acid-Schiff base (PAS), and (**c**) claudin 3-stained colorectal tissues harvested from control (Con), irradiated (IR), and pravastatin-treated IR (IR+PS) mice at 2 and 4 weeks after application of 27 Gy local irradiation. Bar = 100 μm. (**d**) Histological score assessed by the degree of maintenance of epithelial architecture, crypt damage, Paneth cell prominence, and infiltration of inflammatory cells in the lamina propria (0 = none, 1 = mild, 2 = moderate, 3 = high) of the colorectum from Con, IR, and IR+PS groups. Plasma (**e**) diamine oxide (DAO) and (**f**) C-reactive protein (CRP) levels in Con, IR, and IR+PS mice. Data are presented as the mean ± standard error of the mean; *n* = 5 mice per group. ** P* < 0.05 compared to the control; *^#^ P* < 0.05 compared to the IR group.

**Figure 4 ijms-21-01897-f004:**
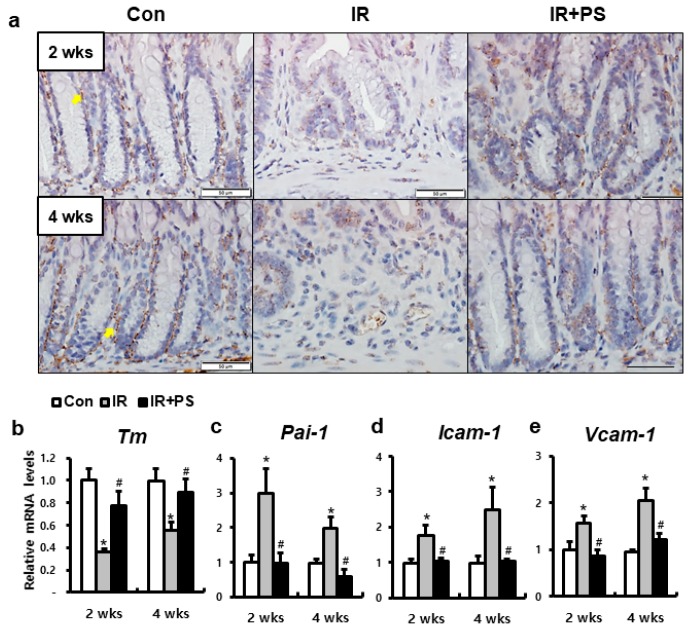
Pravastatin improved thrombomodulin expression and inhibited endothelial adhesion molecules. (**a**) Immunohistochemistry of thrombomodulin (TM) and mRNA levels of (**b**) *Tm*, (**c**) *Pai-1*, (**d**) *Icam-1*, and (**e**) *Vcam-1* in the colorectum of control (Con), irradiated (IR), and pravastatin-treated IR (IR+PS) mice at 2 and 4 weeks after irradiation. Bar = 50 μm. Data are presented as the mean ± standard error of the mean; *n* = 5 mice for each group. ** P* < 0.05 compared to the control; *^#^ P* < 0.05 compared to the IR group.

**Figure 5 ijms-21-01897-f005:**
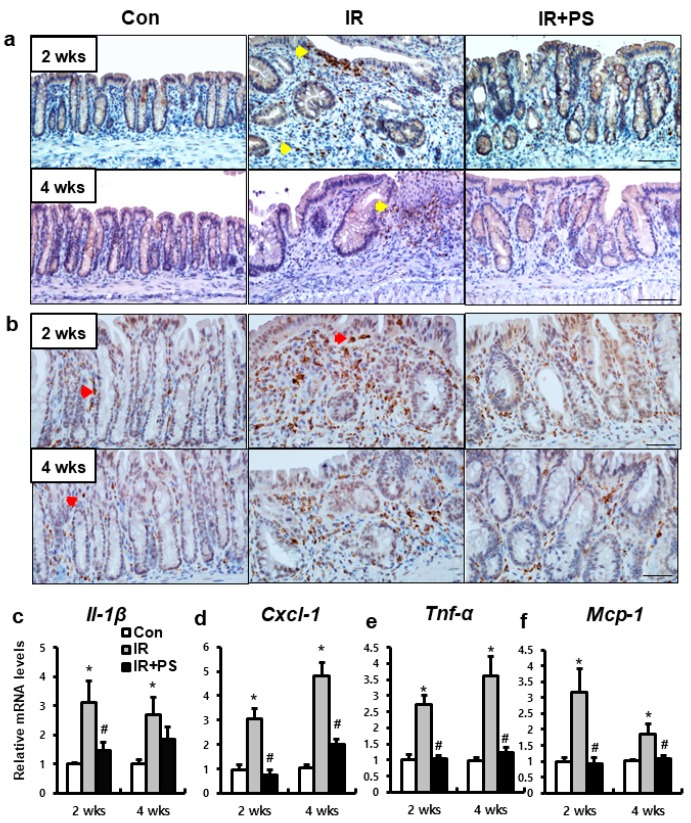
Pravastatin inhibited leukocyte infiltration in colorectal inflammation. Immunohistochemistry of (**a**) myeloperoxidase (Mpo) and (**b**) CD68 for identification of neutrophil and macrophage infiltration, respectively, in the colorectum of control (Con), irradiated (IR), and pravastatin-treated IR (IR+PS) mice at 2 and 4 weeks after radiation exposure. The yellow arrowhead indicates Mpo-positive neutrophil. Bar = 100 μm. The red arrowhead indicates CD68-positive macrophage. Bar = 50 μm. mRNA levels of (**c**) *Il-1β*, (**d**) *Cxcl-1*, (**e**) *Tnf-α*, and (**f**) *Mcp-1* in the colorectum of Con, IR, and IR+PS mice. Data are presented as the mean ± standard error of the mean; *n* = 5 mice for each group. ** P* < 0.05 compared to the control; *^#^ P* < 0.05 compared to the IR group.

**Table 1 ijms-21-01897-t001:** Real-time reverse transcriptase-polymerase chain reaction (RT–PCR) primer sequences.

Species	Primer	Forward (5′–3′)	Reverse (5′–3′)
Human	TM	GGAGCAGCAGTGCGAAGT	GTGGCTGGGAAGTGGAACT
	PAI-1	CCCAGCTCATCAGCCACT	GAGGTCGACTTCAGTCTCCAG
	ICAM-1	GGCCGGCCAGCTTATACAC	TAGACACTTGAGCTCGGGCA
	VCAM-1	TCAGATTGGAGACTCAGTCATGT	ACTCCTCACCTTCCCGCTC
	GAPDH	GGACTCATGACCACAGTCCATGCC	TCAGGGATGACCTTGCCCACAG
Mouse	Il-1 *β*	GCAACTGTTCCTGAACTCA	CTCGGAGCCTGTAGTGCAG
	Cxcl-1	TGAGCTGCGCTGTCAGTGCCT	AGAAGCCAGCGTTCACCAGA
	Tnf-α	GCCTCTTCTCATTCCTGCTT	CACTTGGTGGTTTGCTACGA
	Mcp-1	AGGTCCCTGTCATGCTTCT	CTGCTGGTGATCCTCTTGT
	Tm	TCCCAAGTTTCCATGTTTCC	GCATGAGTTGTGTGCTTCGT
	Pai-1	AGGATCGAGGTAAACGAGAGC	GCGGGCTGAGATGACAAA
	Icam-1	GTGATGCTCAGGTATCCATCCA	CACAGTTCTCAAAGCACAGCG
	Vcam-1	GTTCCAGCGAGGGTCTACC	AACTCTTGGCAAACATTAGGTGT
	b-actin	TCCCTGGAGAAGAGCTATGA	CGATAAAGGAAGGCTGGAA

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
