# Peer review of "Pravastatin Alleviates Radiation Proctitis by Regulating Thrombomodulin in Irradiated Endothelial Cells"

_ijms, 2020, doi:10.3390/ijms21051897_

Round 1

Reviewer 1 Report

The authors have addressed the adverse clinical condition of proctitis following pelvic radiotherapy. They demonstrate a potential protective role of pravastatin against endothelial damage via regulating thrombomodulin expression. However, I have a couple of major suggestions.

The authors seem to have investigated CRP levels to demonstrate increased inflammation, then the anti-inflammatory effect of TM is being investigated by demonstrating its inhibitory action on PAI-1, ICAM-1 and VCAM-1. This does not have a follow through from the first results and it may be misleading to the reader. I suggest the authors should redesign the results section clearly. Although the authors have executed the experiments well, the result section needs a major revision. My suggestion for results section is as follows, Subheading 1 – report the anti-inflammatory effect of pravastatin on inflammatory markers by regulating TM in HUVECs (Section 2.4) Subheading 2 – report that TM improved radiation induced endothelial dysfunction in HUVECs (section 2.5) Following the cell culture results, add few lines to justify these results to be tested on animal model. Subheading 3 – (Establishment of the model) – Report all details about the animals used, and confirm the model development by showing the increase in inflammatory cytokines and others histology (parts of section 2.1 & 2.2). Subheading 4 – demonstrate the improved effect of pravastatin on inflammatory environment in proctitis and the improvement was via accelerated TM expression (parts of 2.1, 2.2 & all of section 2.3)

The authors can redesign to their convenience, however, I strongly recommend to start the results with cell culture data followed by animal data.

Author Response

Reviewer 1

The authors have addressed the adverse clinical condition of proctitis following pelvic radiotherapy. They demonstrate a potential protective role of pravastatin against endothelial damage via regulating thrombomodulin expression. However, I have a couple of major suggestions.

The authors seem to have investigated CRP levels to demonstrate increased inflammation, then the anti-inflammatory effect of TM is being investigated by demonstrating its inhibitory action on PAI-1, ICAM-1 and VCAM-1. This does not have a follow through from the first results and it may be misleading to the reader. I suggest the authors should redesign the results section clearly. Although the authors have executed the experiments well, the result section needs a major revision. My suggestion for results section is as follows, Subheading 1 – report the anti-inflammatory effect of pravastatin on inflammatory markers by regulating TM in HUVECs (Section 2.4) Subheading 2 – report that TM improved radiation induced endothelial dysfunction in HUVECs (section 2.5) Following the cell culture results, add few lines to justify these results to be tested on animal model. Subheading 3 – (Establishment of the model) – Report all details about the animals used, and confirm the model development by showing the increase in inflammatory cytokines and others histology (parts of section 2.1 & 2.2). Subheading 4 – demonstrate the improved effect of pravastatin on inflammatory environment in proctitis and the improvement was via accelerated TM expression (parts of 2.1, 2.2 & all of section 2.3)

The authors can redesign to their convenience, however, I strongly recommend to start the results with cell culture data followed by animal data.

  • We appreciated your comments. To respond your comment, we revised figures and manuscript in the result section.

Reviewer 2 Report

The study confirms that pravastatin can attenuate radiation proctitis by improving endothelial dysfunction. There is several papers dealing with this topic, however, results presented in the reviewed paper indicated that thrombomodulin is a therapeutic target for radiation-induced endothelial injury.

Figure 1D  – results are not clear in panel D. Please specify the description of Y axis. What score is for the control group?

Line 330, we read: “Radiation induces crypt destruction with abscess formation, inflammatory cell infiltration, and epithelial damage, including reduction in tight junction and plasma DAO levels.” – however, in Fig. 1e we observed elevated levels of DAO after IR. Unirradiated control is 55 mU/ml and in IR is 100 mU/ml. – please explain this difference.

Line 364, Were 6 mice per every group: control,  irradiation (IR), and irradiation with pravastatin treatment (IR+PS)? Was it together 18 mice or only 6 for all 3 groups?

Line 396, please specify the manufacturer and Kit number - “Mouse DAO (Cusabio) and CRP (R & D Systems) levels were measured using a commercial enzyme-linked immunosorbent assay (ELISA) kit following the manufacturer’s protocol."

Author Response

Dr. Sibyl Sun

Assigned Editor

International Journal of Molecular Science

March 07, 2020

RE: Ms#IJMS -724473

Dear Dr. Sibyl Sun

   Thank you very much for response, with regard to our manuscript entitle “Pravastatin Alleviates Radiation Proctitis by Regulating Thrombomodulin in Irradiated Endothelial Cells” together with the comments from the reviewers. According to your suggestions, we conducted additional experiments and revised our manuscript.

Our alterations after the reviewers’ comments as follows:

Reviewer 1

The authors have addressed the adverse clinical condition of proctitis following pelvic radiotherapy. They demonstrate a potential protective role of pravastatin against endothelial damage via regulating thrombomodulin expression. However, I have a couple of major suggestions.

The authors seem to have investigated CRP levels to demonstrate increased inflammation, then the anti-inflammatory effect of TM is being investigated by demonstrating its inhibitory action on PAI-1, ICAM-1 and VCAM-1. This does not have a follow through from the first results and it may be misleading to the reader. I suggest the authors should redesign the results section clearly. Although the authors have executed the experiments well, the result section needs a major revision. My suggestion for results section is as follows, Subheading 1 – report the anti-inflammatory effect of pravastatin on inflammatory markers by regulating TM in HUVECs (Section 2.4) Subheading 2 – report that TM improved radiation induced endothelial dysfunction in HUVECs (section 2.5) Following the cell culture results, add few lines to justify these results to be tested on animal model. Subheading 3 – (Establishment of the model) – Report all details about the animals used, and confirm the model development by showing the increase in inflammatory cytokines and others histology (parts of section 2.1 & 2.2). Subheading 4 – demonstrate the improved effect of pravastatin on inflammatory environment in proctitis and the improvement was via accelerated TM expression (parts of 2.1, 2.2 & all of section 2.3)

The authors can redesign to their convenience, however, I strongly recommend to start the results with cell culture data followed by animal data.

  • We appreciated your comments. To respond your comment, we revised figures and manuscript in the result section.

Reviewer 2

The study confirms that pravastatin can attenuate radiation proctitis by improving endothelial dysfunction. There is several papers dealing with this topic, however, results presented in the reviewed paper indicated that thrombomodulin is a therapeutic target for radiation-induced endothelial injury.

Figure 1D – results are not clear in panel D. Please specify the description of Y axis. What score is for the control group?

  • We explained index of histological score in line 450-454.

Line 330, we read: “Radiation induces crypt destruction with abscess formation, inflammatory cell infiltration, and epithelial damage, including reduction in tight junction and plasma DAO levels.” – however, in Fig. 1e we observed elevated levels of DAO after IR. Unirradiated control is 55 mU/ml and in IR is 100 mU/ml. – please explain this difference.

  • Thanks for your comment. We revised the sentence.

Line 364, Were 6 mice per every group: control, irradiation (IR), and irradiation with pravastatin treatment (IR+PS)? Was it together 18 mice or only 6 for all 3 groups?

  • To respond your comments, we added the word (line 436-437).

Line 396, please specify the manufacturer and Kit number - “Mouse DAO (Cusabio) and CRP (R & D Systems) levels were measured using a commercial enzyme-linked immunosorbent assay (ELISA) kit following the manufacturer’s protocol."

  • Thanks for your comment. We revised the sentence (line 467-468).

We are grateful that the manuscript has been improved satisfactory and hope that it would be accepted for publication in the International Journal of Molecular Science.

Very sincerely yours,

Sehwan Shim, DVM, PhD

Senior research

Korea Institute of Radiological and Medical Science

Round 2

Reviewer 2 Report

I accept authors' revision. Thank you.